# Multi-Omics Integration to Reveal the Mechanism of Sericin Inhibiting LPS-Induced Inflammation

**DOI:** 10.3390/ijms24010259

**Published:** 2022-12-23

**Authors:** Yueting Sun, Wenyu Shi, Quan Zhang, Haiqiong Guo, Zhaoming Dong, Ping Zhao, Qingyou Xia

**Affiliations:** 1Biological Science Research Center, Integrative Science Center of Germplasm Creation in Western China (CHONGQING) Science City, Southwest University, Chongqing 400715, China; 2Key Laboratory for Germplasm Creation in Upper Reaches of the Yangtze River, Ministry of Agriculture and Rural Affairs, Chongqing 400715, China; 3Engineering Laboratory of Sericultural and Functional Genome and Biotechnology, Development and Reform Commission, Chongqing 400715, China

**Keywords:** sericin, LPS, inflammation, PRRs signaling pathway, MyD88, NF-κB

## Abstract

Sericin is a natural protein with high application potential, but the research on its efficacy is very limited. In this study, the anti-inflammatory mechanism of sericin protein was investigated. Firstly, the protein composition of sericin extracts was determined by Liquid Chromatography-Tandem Mass Spectrometry (LC-MS/MS). This was then combined with Enzyme-linked Immunosorbent Assay (ELISA) and Quantitative Real-time PCR (qRT-PCR), and it was confirmed that the anti-inflammation ability of sericin was positively correlated with the purity of sericin 1 protein. Finally, RNA-seq was performed to quantify the inhibitory capacity of sericin sample SS2 in LPS-stimulated macrophages. The gene functional annotation showed that SS2 suppressed almost all PRRs signaling pathways activated by lipopolysaccharides (LPS), such as the Toll-like receptors (TLRs) and NOD-like receptors (NLRs) signaling pathways. The expression level of adaptor gene MyD88 and receptor gene NOD1 was significantly down-regulated after SS2 treatment. SS2 also reduced the phosphorylation levels of NF-κB P65, P38, and JNK, thereby reducing the expressions of IL-1β, IL-6, INOS, and other inflammatory cytokines. It was confirmed that sericin inhibited LPS-induced inflammation through MyD88/NF-κB pathway. This finding provides necessary theoretical support for sericin development and application.

## 1. Introduction

Sericin is a natural macromolecular protein, accounting for about 25% of silk weight [1,2]. Sericin contains abundant serine and charged amino acid residues, which may contribute to silk’s stickiness and aid in silk cocoon construction [3]. With the completion of the silkworm genome project and the development of proteomic methods, the molecular mechanism of silk protein synthesis has been revealed [4,5,6]. There are three sericin genes named sericin 1–3 located on chromosome 11, but their expression shows spatiotemporal specificity [7]. Sericin 3 protein is secreted primarily in the outer cocoon coat and tends to be lost before extraction. Therefore, industrially extracted sericin solutions contain more sericin 1 and sericin 2. Besides fibroin and sericin, the sericin layer also contains some relatively small size proteins which are encapsulated in sericin, such as protease inhibitors, serum proteins, and enzymes [8]. Small molecular proteins are extracted along with sericin through the industrial process, and sericin extract is a mixture of these proteins in a broad sense.

Sericin was discharged as waste during the silk degumming process for a long time. During the past several years, various biological activities of sericin have been reported, including use in cell culture [9] and as a health food for patients with diabetes [10,11], antioxidant activities [12,13,14], wound healing effects [15,16,17,18], inhibition of melanin production [19,20,21], reduction in the harmful effects of a high-cholesterol diet [22,23,24], prevention of alcohol-induced liver and stomach damage [25,26], and inhibition of ultraviolet-induced keratinocyte damage [27]. In addition, sericin can be used as a base for anti-inflammatory materials [28], and it is non-allergenic [29]. Some researchers injected sericin into rabbits and sericin antibodies were not triggered, proving sericin has no immunogenicity [30]. However, the mechanism of sericin’s anti-inflammatory action remains unclear.

The innate immune system is the organism’s first line of defense against microorganisms. Viral infection triggers pattern recognition receptors (PRRs) of hosts that can recognize pathogen-associated molecular patterns (PAMPs) or danger-associated molecular patterns (DAMPs) to initiate innate immune responses [31]. Innate immune receptors have been increasingly studied over the past decade with the increase in genome sequencing projects. They are currently divided into five families: (a) Toll-like receptors (TLRs), which are transmembrane proteins whose extramembrane domains are involved in attachment and recognition at the extracellular surface or in endosomes, and are involved in cytoplasmic domains in signal transduction; (b) NOD-like receptors (NLR), which are intracellular cytoplasmic sensors; (c) Retinoic acid-inducible gene (RIG)-I-like receptors (RLRs), which are cytoplasmic sensors that primarily sense viruses’ helicase; (d) C-type lectins (CTLs); and (e) absent-in-melanoma (AIM)-like receptors (ALRs) [32,33]. Among them, the TLR4 is representative of the PRRs. TLRs are type 1 transmembrane proteins transported between the plasma membrane and endosomal vesicles. They are primarily responsible for the detection of PAMPs in the extracellular environment. Gram-negative bacterial outer membrane marker lipopolysaccharides (LPS) is the major PAMP which activates TLR4 [34]. During infection with Gram-negative bacteria, TLR4 responds to LPS present in the tissues and the bloodstream and initiates a pro-inflammatory response that promotes the eradication of invading bacteria [35,36]. TLR4 signals as a dimer and differentially recruits the adaptor proteins myeloid differentiation primary response gene 88 (MyD88) adaptor-like (Mal), also known as TIR domain-containing adaptor protein (TIRAP) and MyD88 [37]. MyD88 is recruited by TIRAP to form an oligomeric signaling scaffold called MyDDosome, which then leads to autophosphorylation of interleukin receptor-associated kinase 4 (IRAK-4) and further activates IRAK-1 to induce TNF receptor-associated factor 6 (TRAF6) interaction with TAK1, TAB1, and TAB2/3 [38]. Afterward, the induced formation of the IKK complex consisting of NEMO (IKKγ), IKKα, and IKKβ leads to the phosphorylation and degradation of IκB [39,40], which ultimately activates the nuclear transcription factor NF-κB, mitogen-activated protein kinase (MAPK) [41], and interferon regulatory factor (IRF). The transcription factors drive the expression of interferons (IFNs), cytokines, and chemokines, and the excessive accumulation of various inflammatory mediators such as nitric oxide (NO), which in turn causes inflammatory responses [31].

Sericin has shown excellent function in many fields, especially materials science and tissue engineering. However, research on the anti-inflammation effect of sericin has not formed a system. In this study, the high-throughput proteomics method was used to accurately quantify the protein abundance of sericin extracted by high temperature and high pressure. Combining molecular biology approaches, a quantitative relationship between sericin and its ability to resist LPS-induced inflammation was found. Based on RNA-seq, the mechanism of sericin to reduce LPS-induced inflammation was elucidated on a genome-wide scale. It was found that sericin could inhibit the expression of Myd88 (the adapter protein of TLR4), which inhibited the NF-κB phosphorylation and decreased downstream inflammatory cytokine expression. Finally, sericin alleviated the inflammatory response. This paper provides a partial basis for sericin efficacy evaluation and theoretical support for regenerating industrial silk resources.

## 2. Results

### 2.1. The Physicochemical Properties of Sericin

Five sericin samples from different sources were selected for parallel experiments, including two sericin samples, SS1 and SS2, extracted in our laboratory and three commercially available sericin samples, S1, S2, and S3 (Appendix A). SS1 and SS2 were treated with high temperature and high pressure for 1 h and 1.5 h, respectively. The molecular weight of SS1 (80–300 kDa) and SS2 (20–50 kDa) was detected by SDS-PAGE. The main band of the sericin decreased with the increase in heating time and reached the critical value after 1.5 h (Figure 1A).

The molecule weight (Appendix A), appearance, odor, and pH value of the five sericin samples were counted (Table 1, Figure 1B). All sericin samples appeared to be acidic solutions (pH values from 4.53 to 5.60). All sericin samples had almost no aerobic bacteria (Table 1). All samples had a relative nitrogen content between 14% and 18%, which meant the protein content of the sample was close to 100% (F = 6.25) (Figure 1C).

### 2.2. Protein Composition Analysis of Sericin Samples

The protein components of the five sericin samples were detected by Liquid Chromatography-Tandem Mass Spectrometry (LC-MS/MS) [42]. The protein composition of S1, S3, SS1, and SS2 was relatively similar (Figure 2A, Appendix A). In total, 315 proteins were identified in the five sericin samples. The molecule weight of sericin 1 was the highest among the 315 proteins (Figure 2B). Sericin 1 (NP_001037506.1, NCBI) and sericin 2 (NP_001166287.1, NCBI) were detected in all five samples. The purity of the sericin protein in SS1 (93.97%) and SS2 (94.71%) was the highest and was much higher than that in S1 (66.73%), S2 (5.37%), S3 (30.78%). Furthermore, there were also non-sericin proteins detected, such as enzymes (XP_021208623.1 et al. NCBI), protease inhibitors (BGIBMGA009092 et al. SilkDB2.0), and unannotated proteins (BGIBMGA000013 et al. silkDB2.0) (Figure 2B, Table 2). These components were consistent with those shown in literature reports [2,8].

Even if the composition of each protein sample was not precisely the same, they were mainly composed of sericin, enzyme, protease inhibitor, and unannotated protein parts. Interestingly, the composition of several parts varied. S2 contained more unannotated proteins, and S3 contained more enzymes (Table 2). It was speculated that the difference was related to the silkworm cocoon type and the sericin production process. The protein abundance of S1, S3, SS1, and SS2 was highly correlated. SS1 and SS2 showed the highest correlation (r_pearson_ = 0.9936, *p* < 0.001) (Figure 2C), which was consistent with the fact that the cocoon varieties and sources of SS1 and SS2 were precisely the same. PCA analysis showed that S2 was different from the other four samples (Figure 2D). The reason was the more unannotated protein and fewer sericin components in S2 (Table 2 and Appendix A). Based on this, we wanted to know the differences in biological function among sericin samples with different compositions.

### 2.3. Sericin Is a Multifunctional Biomacromolecule

To explore the biological function of sericin, we first detected the cytotoxicity of SS1 and SS2, which showed the highest sericin content. The IC_50_ of SS1 and SS2 was 46.29 mg/mL and 48.23 mg/mL, respectively, and both showed no cytotoxicity (Figure 3A,B). The IC_25_ (13.5 mg/mL) was determined as the highest concentration in subsequent experiments on the cellular level. The ocular irritation and corrosion of sericin were detected by a hen’s egg test on the chorioallantoic membrane (HET-CAM) experiment [43]. Different concentrations of SS1 and SS2 solution (5%, 7.5%, 10%) were prepared for detection, 10% of which was almost the limited concentration of sericin before it became gel form. The results showed SS1 and SS2 did not cause hemorrhage, coagulation, or blood vessel lysis of the CAM (Figure 3C,D). The HET-CAM endpoint score (ES) was significantly lower than the positive control (Table 3). The above results proved sericin was a safe and non-toxic substance.

With the deepening of the research, we found that sericin was a multifunctional biomacromolecule. Sericin appeared to promote cell proliferation, anti-inflammation, and antioxidation (Appendix A). The micrograph showed that when the medium contained sericin, NIH 3T3 cell density was higher after growth for 24 h (Figure 3E). The CCK-8 results showed that adding S1, SS1, and SS2 to the culture medium could increase cell survival rate (cell proliferation rate: 28.36%, 37.01%, 31.02%) (Figure 3F). At the same time, S2 and S3 could not improve cell survival rates. We speculated that this might be because sericin proteins in these two samples were not as abundant as in others.

Sericin also showed a strong ability to decrease the cytokines induced by LPS in RAW 264.7 cells. The decrease in inflammatory cytokines was synchronously detected by real-time quantitative PCR and ELISA. SS1 and SS2 showed a suppressive effect on the inflammatory cytokine TNF-α in both mRNA and protein levels (Figure 3G). S1, S3, SS1, and SS2 showed the ability to inhibit the production of interleukin-6 (IL-6) (Figure 3H), and S1, SS1, and SS2 showed the inhibition of interleukin-1β (IL-1β) (Figure 3I) in both mRNA and protein level. S3 was labile in suppressing the inflammatory cytokines. S2 showed almost no ability to suppress the cytokines. In summary, the anti-inflammatory effects of five sericin samples were ranked as SS2 > SS1 > S1 > S3 > S2.

It was worth noting that the anti-inflammatory capacity of S2 was the worst among the five samples. Meanwhile, the LC-MS/MS analysis reported that the component of S2 was the most unusual (Figure 2A), including more than 93% of unannotated protein and lower than 5% of sericin 1 protein (Table 2). Correspondingly, the SS2 and SS1 samples with the highest sericin 1 protein content had the best anti-inflammatory effect. This suggested that sericin, especially sericin 1, might be the key to the anti-inflammatory effect of the samples.

### 2.4. Sericin Suppressed LPS-Induced Immune Responses through the Pathogen Recognition Receptors (PRRs) Signaling Pathways

Since the sericin protein content of SS2 was more than 94.7%, SS2 might be the best sample to verify the anti-inflammatory function of sericin. To define the anti-inflammatory mechanism of sericin protein, four kinds of treatments were designed for RAW 264.7 cells, including the wild-type cells (WT), the SS2-treated group (S), the LPS-infected group (L), and the SS2-added group after LPS infection (LS) for 4 h. RNA-seq was performed on the four groups with three subsequent repeats (Figure 4A).

The RNA-seq data showed a high correlation among each replicate. Gene expression level between L and WT showed a low correlation (Figure 4B), which indicated that LPS stimulated the RAW 264.7 cells. The S and LS groups appeared to correlate highly (Figure 4B). The correlation between LS and S was more significant than that with L, which indicated that after adding SS2 to the L group, the stimulation activated by LPS was inhibited. The gene expression level also appeared in the same pattern. There were many more (1133) Different Expressed Genes (DEGs) between WT and L, and fewer DEGs (9) between S and LS (Figure 4C and Appendix A). All the above results proved that sericin inhibited the LPS-aroused stimulation. Therefore, follow-up analysis on the specific gene function needed to begin.

Since there were few DEGs between L and LS, Gene Set Enrichment Analysis (GSEA) was suitable for annotating Gene Ontology (GO) and Kyoto Encyclopedia of Genes and Genomes (KEGG) enrichment [44], of which GSEA was based on the complete gene expression profile but not the DEGs. GO enrichment results reported that LPS aroused many inflammation-related biological responses. Here we visualized the top 10 biological processes. Many inflammation responses were up-regulated after LPS was added to cells, such as cytokines production, defense response, response to biotic stimulus, and response to the bacterium (Figure 4D, Appendix A), consistent with previous reports [35,45,46]. Amazingly, most of the biological processes mentioned above were down-regulated after sericin protein was added to the LPS-infected RAW 264.7 cells; typical examples include the response to interferon, virus, and bacterium (Figure 4E).

To identify the certain signaling pathway that suppressed the inflammatory reaction, we performed *gseaKEGG* with an R package clusterProfiler [47]. The KEGG enrichment analysis showed that LPS activated the NLRs signaling pathway (mmu04621), NF-κB pathway (mmu04064), virus infection pathways (mmu05168, mmu05171, mmu05169, mmu05163, mmu167 et al.), TNF signaling pathway (mmu04668), PI3K-Akt signaling pathway (mmu04151), TLRs signaling pathway (mmu04620), JAK-STAT signaling pathway (mmu04630), and so on (Figure 5A, Appendix A). These pathways were mainly associated with immune inflammation, oxidative stress, and metabolism. This indicated that LPS stimulated the macrophages and triggered inflammatory responses [35,46].

After SS2 was added, many inflammation-associated pathways were suppressed, including TLRs signaling pathway (mmu04620), cytokine-cytokine receptor interaction pathway (mmu04060), JAK-STAT signaling pathway (mmu04630), virus infection pathways (mmu05171, mmu05165, mmu05203, mmu05161), TNF signaling pathway (mmu04668), NLRs signaling pathway (mmu04621), et.al. It was found that PRRs-related signaling pathways were much higher in confidence, such as TLRs and NLRs signaling pathways (Figure 5B and Appendix A). These results suggested that sericin could suppress LPS-induced inflammation through the PRRs signaling pathways.

### 2.5. Molecular Mechanism of Sericin Inhibiting LPS-Induced Inflammation

Signaling pathways were visualized to detect the changes in gene expression level immediately by ClusterProfiler and Pathview [47,48] (Figure 6A). The 20 most reliable pathways and the enriched genes are visualized. PRRs-related signaling pathways were increased in L vs. WT and decreased in LS vs. L, including TLRs signaling pathway, NLRs signaling pathway, and RLRs signaling pathway (Figure 6A,B).

After LPS stimulation, *MyD88* and *TIRAP* were up-regulated and activated the subsequent inflammatory response. The expression level of *NF-κB* downstream genes was distinctly changed after LPS treatment (Figure 6B and Figure 7A,B). Subsequently, NF-*κ*B entered the nucleus and initiated the transcription and expression of pro-inflammatory genes, including the critical tumor necrosis factor receptor-related factors *TRAF3, TRAF6*, mitogen-activated protein kinase kinase kinase (MAP3K) family member *TAK1* and its binding protein *TAB1*, and interferon regulatory factor *IRF7*. These genes were involved in the development of inflammation after LPS stimulated and ultimately triggered the expression of cytokine genes *IL-1β*, *IL-6*, *IL-18*, inflammatory mediator genes *iNOS*, interferon genes *IFNGR2*, and chemokine genes *CCL5* (Figure 6B and Figure 7B). In addition to the TLRs signaling pathway, *NOD1*, *NOD2* gene, and NLRs signaling pathways were captured. Based on previous reports, we hypothesized that *NF-κB* promoted the activation of the *NOD1* and *NOD2* genes, which activated the NLRs signaling pathway, further forming the inflammatory cascade amplification [49] (Figure 6B).

After SS2 treatment, the expression of most genes was significantly down-regulated, such as *MyD88*, *TIRAP, TAB2, TBK1, IKKβ, NOD1, RIKP2, IL-1β*, *IL-6*, *IL-18, CCL2,* and *CCL5*. As a result, the enriched signaling pathways showed a downward trend (Figure 6A). The signaling pathways with the most significant differences in gene expression were enriched. This suggested that LPS activated inflammation pathways and made inflammation cytokines, and sericin inhibited inflammation products through *MyD88* and *TIRAP* (Figure 6B). Most of the PRRs-related genes captured by RNA-seq showed a trend of L > WT, L > LS, and LS > S. STEM clustering was used to cluster genes stimulated by LPS but inhibited by sericin [50]. There were about 750 genes showing the same expression tendency (Figure 7A, Appendix A), and the profiles of representative genes expression were established (Figure 7B and Appendix A). In summary, after LPS stimulated macrophages, the PRRs signaling pathways were activated, while after adding SS2, the inflammatory pathways were suppressed.

To ensure and complete the reliability of the RNA-seq, several vital proteins were verified by WB. Activating the critical adaptor MyD88 was an essential step of signal transduction after LPS stimulation [46]. We analyzed the abundance of MyD88 in RAW 264.7 cells. The results showed that the abundance of MyD88 increased after LPS stimulation but decreased significantly after adding different concentrations of SS2. The inhibitory effect of sericin was concentration-dependent (Figure 7C and Appendix A). To determine whether the NF-*κ*B and MAPKs had been activated, we detected the protein abundance and phosphorylation of NF-κB P65, kinases JNK and P38 of the MAPK signaling pathway. The results showed that their phosphorylation was suppressed after SS2 addition (Figure 7C and Appendix A). Finally, cytokines IL-1β and IL-6 were detected at mRNA and protein levels (Figure 7D). The expression levels of IL-1β and IL-6 were significantly increased after LPS stimulation, but after adding SS2, their expressions decreased with the increase in SS2 concentration (* *p* < 0.05, ** *p* < 0.01, *** *p* < 0.001). The protein abundance trend was very similar to that captured by RNA-seq.

## 3. Discussion

Silkworms have been domesticated for over 5000 years as silk-production insects. Sericin is one of the main components of natural silk, and is synthesized and generated in the silkworm’s silk gland. The traditional silk industry discharges sericin as industrial waste, polluting water and resources. In recent years, sericin has made great strides in regenerative medicine, tumor diagnostics, and materials science as a natural polymer material. To advocate for the green industry and turn waste into treasure, we used an industrial scale-up plan to prepare sericins with different molecular weights. Furthermore, we found three samples on the market for parallel research. Taking advantage of the insolubility of silk fibroin, boiling silk in water is a standard method for sericin extraction. Acids, alkalis, enzymes, and high pressure can increase efficiency. The extracted sericin is in the form of a suspension and can also be made into powder.

Nevertheless, other water-soluble proteins are usually extracted during the extraction. There are hundreds of proteins in silkworm cocoons, of which fibroin and sericin are the most abundant. In the sericin layer, there are also protease inhibitors, serotonin, enzymes, and unannotated proteins. According to the LC-MS/MS analysis, the protein composition of the five sericin samples conformed to the general rules. We identified multiple proteins in all five sericin samples, including sericin 1, sericin 2, fibroin, enzymes, and unannotated proteins. By comparing several samples, we found that the protein form could affect the physicochemical properties and biological activity of sericin, especially after a period of storage. It could be seen that liquid sericin was more prone to discoloration and degradation. There might be two reasons for the brown or black color of sericin. First, phenol oxidase in silk could catalyze the production of DOPA from tyrosine, which produces melanin [8]. Second, the Maillard reaction might occur between sericins and carbohydrates in the silk [51]. The exceptionally dark color of some samples might be related to the incomplete sequestration of carbohydrates during the extraction process. In addition, liquid sericin samples were more prone to unpleasant odors. It suggested sericin was best stored and transported in dehydrated form. Some sericin samples showed better anti-inflammation effects. Since five sericin samples had no aerobic bacteria contamination (Table 1), the different ability of inflammation inhibition was not microbe-driven. Combined with proteomic analysis, we believed that sericin purity was the key to anti-inflammation. SS2 was the most purified sericin sample and exhibited the best anti-inflammatory effect. Therefore, SS2 was used to investigate the anti-inflammatory mechanism of sericin.

LPS is the cell wall component of Gram-negative bacteria. LPS can stimulate cell membrane receptor TLR4, activate the innate immune pathway, and trigger a series of inflammatory responses, seriously endangering the health of organisms [52,53]. The main treatment methods for the inflammation caused by LPS are antibiotics, hormones, and other conventional methods, which can cause some adverse effects on patients, such as drug resistance, allergy, and metabolism disorders. Therefore, it is of great significance to develop new LPS control strategies. In this study, natural sericin was added to macrophages after LPS infection for 4 h, and cells were cultured for 12 h. The changes in gene expression levels were detected on a genome-wide scale. The cytokines and interferon genes such as *IL-1β*, *IL6*, *IL18*, *CCL*2, *CCL5,* and *IFN* were significantly down-regulated, which preliminarily indicated that sericin could induce the reduction of inflammation. In the upstream, the phosphorylation level of NF-*κ*B P65 was significantly down-regulated, and the phosphorylation levels of the kinases JNK and P38 from the MAPK family decreased accordingly. This indicated that sericin inhibited the NF-*κ*B signaling pathway activated by LPS. Through the KEGG annotation, we found that multiple up-regulated pathways induced by LPS were significantly inhibited, which were all involved in the NF-*κ*B signal transduction. Consistent with previous findings, the classical TLR4 signaling pathway was significantly inhibited. LPS stimulates the TLR4/MD2 complex to recruit MyD88 in large quantities, which presents signals to NF-*κ*B, resulting in its phosphorylation into the nucleus to regulate downstream inflammation. In this study, MyD88 in macrophages was significantly up-regulated after stimulation with LPS, while it was significantly down-regulated after sericin was added. The phosphorylation level of NF-*κ*B P65 also showed a similar trend. So far, we can preliminarily determine that sericin can inhibit cellular inflammation caused by LPS, which functions through the MyD88/NF-κB signaling pathway.

In addition to the TLRs signaling pathway, RNA-seq captured many PRR signaling pathways. Sericin suppressed the various PRR gene expression, such as *NOD1*, *NOD2*, and *RIG-I*. The functional annotation results showed that the PRRs-related signaling pathways were up-regulated after LPS stimulation and down-regulated after sericin addition, especially the inflammatory cytokines and mediators. On the one hand, this showed that LPS stimulated macrophages and activated the innate immune response. On the other hand, one of the characteristics of LPS is causing the inflammatory cascade. Several PRR signaling pathways correlated with each other, eliciting an inflammatory cascade. For example, the NLRs signaling pathway was captured as highly reliable. *NOD1* expression showed a consistent trend with genes in the TLRs signaling pathway. It was reported that LPS could activate the *NOD1* gene by activating NF-*κ*B and inflammatory factors. *NOD1* stimulates NF-*κ*B again and cascading amplifies the inflammatory responses [54]. In this paper, the expression level of *NOD1* was down-regulated after the addition of sericin. We speculated that this was due to the inhibition of MyD88/NF-κB P65, which reduced the production of inflammatory factors and mediators, thereby decreasing the expression of *NOD1*.

Last but not least, we had the following hypotheses regarding the specific mechanism of sericin as an LPS inhibitor. As is known, before LPS induces inflammatory responses, the organisms take a series of countermeasures. First, lipopolysaccharide-binding proteins (LBP) bind to LPS micelles, convert them into monomers, and concentrate them near cell membrane receptors. The LPS monomer is transferred through CD14 to the TLR4/MD-2 complex [52]. CD14 is a glycosyl phosphatidylinositol (GPI)–the anchored protein that transfers LPS to the TLR4/MD2 complex. In the plasma membrane, TLR4 interacts with the adaptor protein MyD88 to further trigger an intracellular inflammatory response [37]. An existing Eritoran inhibitor inhibits LPS through competitive binding with MD2 [53]. Sericin had a similar inhibitory effect. Here, sericin significantly down-regulated the expression of MyD88 and downstream inflammation genes. Therefore, we believed the target of sericin was upstream of MyD88. However, no adequate information was captured on LBP, CD14, TLR4, and MD2. Although sericin had a dose-dependent inhibitory effect on LPS, we could not confirm the specific target. We will focus on this in our future research.

Sericin inhibited LPS-induced MyD88 recruitment and NF-*κ*B activation, regulated downstream inflammatory cytokines, and inhibited activation of PRR signaling pathways. Sericin has potential application prospects in the medicine, food, and cosmetic fields. At the same time, the research team is conducting in-depth research on the specific anti-inflammatory mechanism of sericin, including receptor recognition patterns and core function structures, and the relevant work is steadily advancing.

## 4. Materials and Methods

### 4.1. Materials

Silkworm cocoons used were practical varieties purchased from Chongqing Sericulture Science and Technology Research Institute. LPS were from Sigma. Human immortalized keratinocytes (HaCaT) cells, mouse embryo fibroblast (NIH 3T3) cells, leukemia cells in mouse macrophage (RAW 264.7) cells, and human embryonic kidney (HEK 293) cell lines were from China Center for Type Culture Collection. Dulbecco modified eagle medium (DMEM) and fetal bovine serum (FBS) were from Gibco (Gaithersburg, MD, USA). Milli-Q water was used in all experiments. All other chemicals were of analytical grade or better.

### 4.2. Sericin Extract

Silk cocoons were washed and added to distilled water at a ratio of 1:20. Sericin was extracted at 120 °C for 1 h, 1.5 h, and 2.25 h, and separated by a sifter. The extracted sericins were spray-dried into powder (YAMATO, ALPS-City Yamanashi, Japan) and stored at −20 °C.

### 4.3. Cell Counting Kit-8 (CCK- 8) Assay

NIH 3T3 and HACAT cells in the logarithmic phase were inoculated in 24 well plates at a density of 1 × 10^6^ cells/mL and cultured at 37 °C in a 5% CO_2_ incubator until the cells were 80% confluence. Subsequently, the cells were treated with samples for 24 h. CCK-8 (Yeasen, Shanghai, China) was added to each well, and the cells were incubated for an additional 1–4 h. The optical density (OD) was measured at 450 nm using a microplate reader (Synergy™ H4. Minneapolis, MN, USA), and cell viability was calculated.

### 4.4. Ocular Irritant and Corrosive HET-CAM Test

SPF chicken embryos of Bai Laihang were selected, each weighing 50–60 g. Room temperature 20~25 °C, relative humidity 45~70%. Incubation temperature 37.5 °C ± 0.5 °C, relative humidity 55~70%, turntable frequency 3 times/h~6 times/h. When hatching to 9 days of age, the embryos were inspected with an egg lamp, and unfertilized, defective, broken, or thin-shelled chicken embryos were discarded. Marked the air chamber position on the surface of the eggshell and peeled off the marked part of the eggshell with dental serrated forceps, exposing the white egg film. Added 0.9% NaCl solution to moisten the egg film and poured out the saline. Carefully removed the inner membrane with forceps to ensure that the chorioallantoic membrane (CAM) was not damaged. A quantity of 0.3mL of sericin solution was dripped on the surface of CAM, and after 5 min, whether there was any adverse reaction in CAM was recorded and scored. It was rated as 0 points (none), 1 point (mild), 2 points (moderate), and 3 points (severe), according to the severity of bleeding, coagulation, and vascular melting.

### 4.5. Luciferase Assay

The transfected cells were seeded in 6-well plates and treated with different proteins for 4 h, and then LPS was added for 12 h. Performed the luciferase text according to the instructions (Promega. Madison, WI, USA) and tested the luminous intensity on the microplate reader (Synergy™ H4. Minneapolis, MN, USA).

### 4.6. ELISA Assay

RAW 264.7 cells at a density of 1 × 10^6^ cells per ml were seeded into 24-well plates, incubated with sericins for 4 h, and then stimulated with 200 ng/mL LPS for 24 h. Cell supernatant was collected by centrifugation at 3000 rpm. It was assayed for quantitation of pro-inflammatory cytokines tumor necrosis factor-α (TNF-α), interleukin-1β (IL-1β), and interleukin-6 (IL-6), using a mouse ELISA kit according to the manufacturer’s instructions (R&D System. Minneapolis, MN, USA). Dexamethasone (DXMS) was used as a positive control.

### 4.7. Quantitative Real-Time PCR

The total mRNA of RAW264.7 cells was isolated by Trizol (Invitrogen. San Diego, CA USA) and reversely transcribed to complementary DNA (cDNA) at 42 °C (E047-01B), and NovoStart^®^SYBR qPCR SuperMix plus (E096) was used as the fluorescent dye (Novoprotein. Shanghai, China). The relative mRNA level was normalized with GAPDH. DNA sequence synthesis was completed in Tsingke Co, Ltd., Beijing, China. DXMS was used as a positive control. Primers were detected by AGE, staining with gelRed dye. See Appendix A for details of primers.

### 4.8. Liquid Chromatography-Tandem Mass Spectrometry (LC-MS/MS)

Protein was digested according to the protocol described as the filter-aided sample preparation method. Briefly, the protein was placed in an ultrafiltration tube (MWCO 3000, Millipore, Burlington, MA, USA) and reduced with 10 mm dithiothreitol (DTT) for 120 min at 37 °C, and then alkylated with 50 mm iodoacetamide (IAA) for 60 min in the dark. After washing three times with 8 M urea and then four times with 50 mm NH_4_HCO_3_, proteins were incubated with trypsin (1 μg/50 μg protein) for 36 h at 37 °C in 150 μL 50 mm NH_4_HCO_3_. Tryptic peptides were concentrated and resuspended in 0.1% formic acid and separated by using the Thermo Fisher Scientific (Springs, CO, USA) EASY-nlc 1000 system and EASY-Spray column (C18, 2 μm, 100 Å, 75 μm × 15 cm) with a 2∼95% acetonitrile gradient in 0.1% formic acid over 140 min at a flow rate of 300 nL/min. The separated peptides were analyzed using a Thermo Scientific Q Exactive mass spectrometer operating in data-dependent mode. Up to 10 of the most abundant isotope patterns with charge ≥2 from an initial survey scan were automatically selected for fragmentation by higher energy collisional dissociation with normalized collision energies of 27%. The maximum ion injection times for the survey and MS/MS scans were 20 and 60 ms, respectively. The ion target value for both scan modes was set to 1E6. A dynamic exclusion of ions sequenced previously within the 20 s was applied.

### 4.9. RNA-Seq

Total RNA was isolated using the SV Total RNA Isolate System (Z3100; Promega, Madison, WI, USA) according to the manufacturer’s instructions. All RNAs were screened using an Agilent 2100 Bioanalyzer (Agilent Technologies, Santa Clara, CA, USA) to ensure the sample quality was sufficiently good for RNA-seq library preparation. The rRNA was cleaned using RNAClean XP beads (Illumina. San Diego, CA, USA) and subjected to reverse transcription to obtain cDNAs. After adenylating the three’ ends and ligating the adapters, the library was enriched by a polymerase chain reaction. The average read size of the library was 260 bp, and more than 1 × 10^6^ aligned reads were obtained. RNA-seq data were analyzed using standard methods. The quality of the raw and processed reads was evaluated using FastQC (Version 0.11.1). PolyA tails were filtered by fqtrim (Version 0.93). Low-quality reads were removed with Trimmomatic. Clean reads were aligned to the mouse reference genome (mm10, NCBI) with bowtie2 (Version 2.4.5). Differentially expressed genes were detected by RSEM (Version 1.2.29).

### 4.10. Protein Identification

The resulting raw MS data were analyzed with maxQuant software (version 1.6.0.1). Peptide searches were performed with the Andromeda search algorithms [42]. The search parameters for protein identification specified the initial precursor and fragment mass tolerances of 6 and 20 ppm, respectively. Carbamidomethylation of cysteine was set as a fixed modification, and N-terminal protein acetylation and methionine oxidation were set as variable modifications. The minimal peptide length was set to six amino acids, and up to two miscleavages were allowed. The false discovery rate was set to 0.01 for both peptides and proteins. All common contaminants and reverse hits were removed.

### 4.11. SDS-PAGE

Sericins with the same solid content were prepared and loading buffers were added, followed by boiling for 5 min. Electrophoresis was performed in 12% gel (Thermo Fisher Scientific. Springs, CO, USA). The gel was dyed with Coomassie Blue Dye and canned with a Patch clamp system (Canon. Tokyo, Japan).

### 4.12. Western Blot

After SDS-PAGE, the gel was electro-transferred onto PVDF membranes. The membranes were blocked with 5% BSA and incubated with the primary antibody overnight, followed by washing with TBST and incubation with the secondary antibody for 2 h. The signals were detected by ECL Reagent (Thermo Scientific. Springs, CO, USA).

### 4.13. Aerobic Bacterial Count Assay

The 1 g (1 mL) sericins were diluted 100 times and added to lecithin and tween 80 agar medium (Solarbio Life Science. Beijing, China). Cultured in a 36 °C ± 1 °C incubator for 48 h ± 2 h and counted.

### 4.14. Solid Content/Protein Content Assay

The sericin materials were dried to a constant weight, and the solid content ratio was calculated by the difference in mass before and after drying. The sericin materials were carbonized (Haineng SH40F. Jinan, China) when the initial temperature of the controlled-temperature furnace reached 200 °C. A gradient temperature was used to heat 100 °C every half hour until it reached 450 °C. Protein content was determined with a Kjeldahl device (Haineng K9860. Jinan, China).

### 4.15. Statistical Analysis

Data were presented as mean ± SD. Two-tailed Student’s *t*-tests were performed for statistical analysis. *p* < 0.05 was considered to be statistically significant.

## Figures and Tables

**Figure 1 ijms-24-00259-f001:**
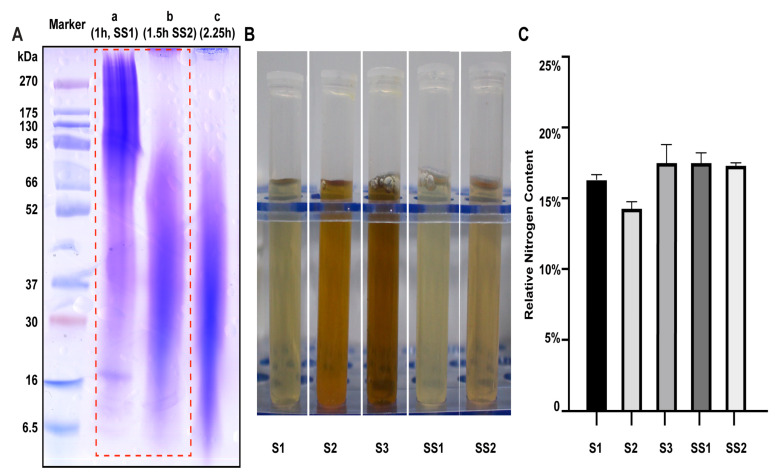
Properties of sericin samples. (**A**) The molecular weight of sericin obtained through different extraction times was different. (**B**) The appearance of sericin solution (13.5 mg/mL). (**C**) The relative nitrogen content of sericin samples.

**Figure 2 ijms-24-00259-f002:**
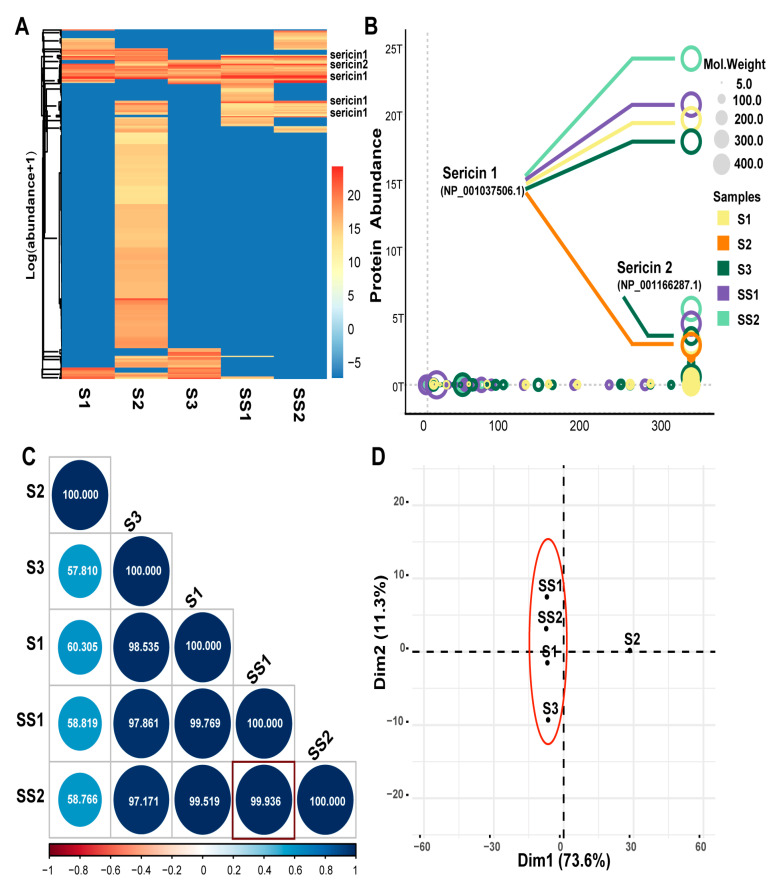
LC-MS/MS analysis of sericin samples. (**A**) The heatmap of protein abundance of the five samples. The protein abundance value was normalized to log (abundance + 1). (**B**) Bubble chart of protein abundance. The color indicates different samples. Bubble size indicates the molecular weight of the protein. The score indicates a comprehensive score of LC-MS/MS data. (**C**) Correlation analysis of protein abundance. Blue indicates a positive correlation and red indicates a negative correlation. The circle size is inversely proportional to the *p*-value. The number in the circle indicates their similarity. SS1 and SS2 show the strongest correlation (r_Pearson_ = 0.99936, *p* < 0.001). (**D**) Principal Component Analysis (PCA) of sericin samples. PC1(73.6%) and PC2 (11.3%) were the top 2 principal components.

**Figure 3 ijms-24-00259-f003:**
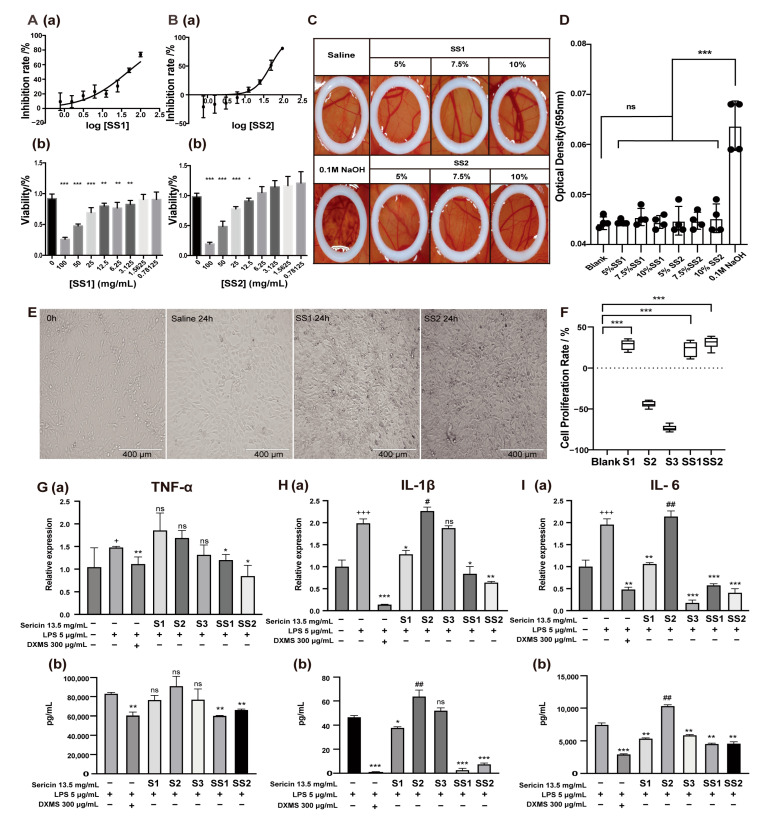
The multiple effects of sericin. (**A**,**B**) The IC_50_ analysis (**a**) and the cytotoxicity (**b**) of SS1 (**A**) and SS2 (**B**). Various concentrations of samples (dilution factor: 1.41) were added to NIH 3T3 cells in the logarithmic growth phase and cell viability was measured with a CCK-8 kit; the IC_50_ of these was calculated. * *p* < 0.05, ** *p* < 0.01, *** *p* < 0.001. n = 6. (**C**,**D**) Stimulation of sericin on CAM. The pictures (**C**) and OD analysis (**D**) of CAM treated with different concentrations of sericin. Sericin showed no irritation to CAM. (**E**) Cell status after treatment with SS1 and SS2 for 24 h. Both SS1 and SS2 treatments (13.5 mg/mL) increased the number of NIH 3T3 cells. (**F**) Cell proliferation rate after 13.5 mg/mL sericin samples treatment for 24 h. *** *p* < 0.001. n = 6. (**G**–**I**) Effect of sericin samples on TNF-α (**G**), IL-1β (**H**), and IL-6 (**I**) produced by LPS-induced macrophages. The levels of inflammatory cytokines were measured by qRT-PCR (**a**) and ELISA (**b**). + Relative to no-treated cells, * and # relative to LPS-treated cells. +Indicated promotion, + *p* < 0.05, +++ *p* < 0.001. * Indicated suppression, * *p* < 0.05, ** *p* < 0.01, *** *p* < 0.001. # Indicated promotion, ## *p* < 0.01. n = 3.

**Figure 4 ijms-24-00259-f004:**
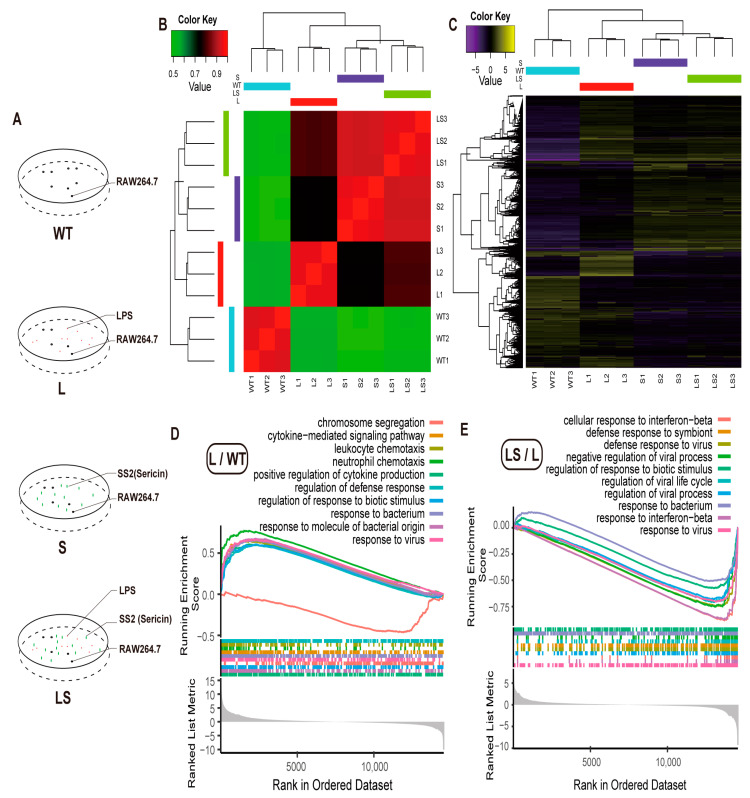
GO analysis. (**A**) Experimental design. (**B**) Heatmap of sample correlation analysis. (**C**) Heatmap of differential gene expression. (**D**,**E**) Gene Ontology (GO) annotation of the DEGs of L/WT (**D**) and LS/L (**E**).

**Figure 5 ijms-24-00259-f005:**
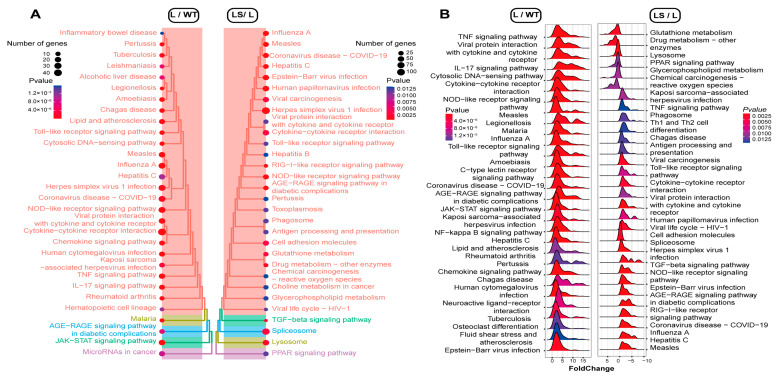
The *gseaKEGG* annotation. The *p*-value decreased from blue to red. (**A**) Treeplots of the different pathways among L, WT, and LS. The pathways were clustered by *p*-value. (**B**) Ridgeplots of the different pathways among L, WT, and LS. The up/down-regulation was marked with an *x*-axis (X > 0, up-regulated; X < 0, down-regulated).

**Figure 6 ijms-24-00259-f006:**
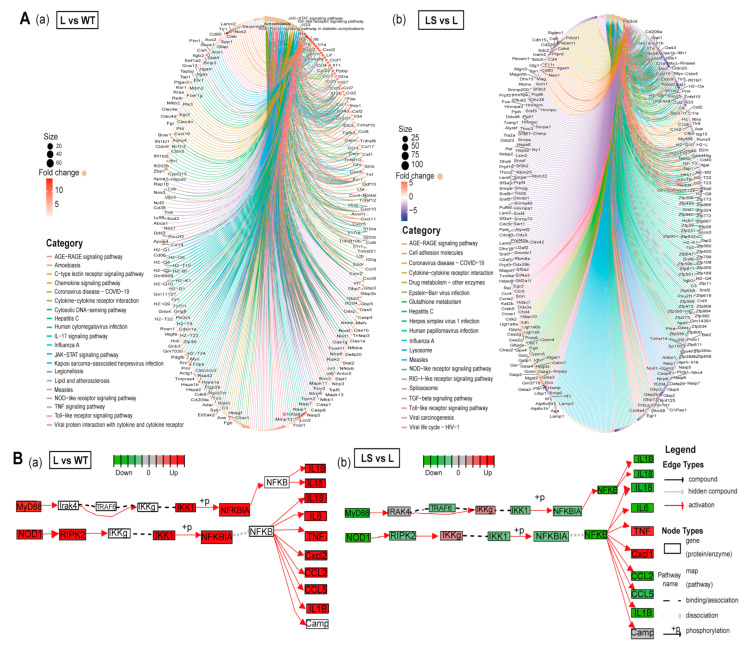
The anti-inflammatory mechanism of sericin SS2. (**A**) Gene regulation in the top 20 different pathways. The fold change decreases from red to purple. (**a**) L/WT. (**b**) LS/L. (**B**) The flow chart of differential gene expression level in MyD88/NF-κB pathway among different samples. (**a**) L/WT. (**b**) LS/L. Green represents down-regulation and red represents up-regulation.

**Figure 7 ijms-24-00259-f007:**
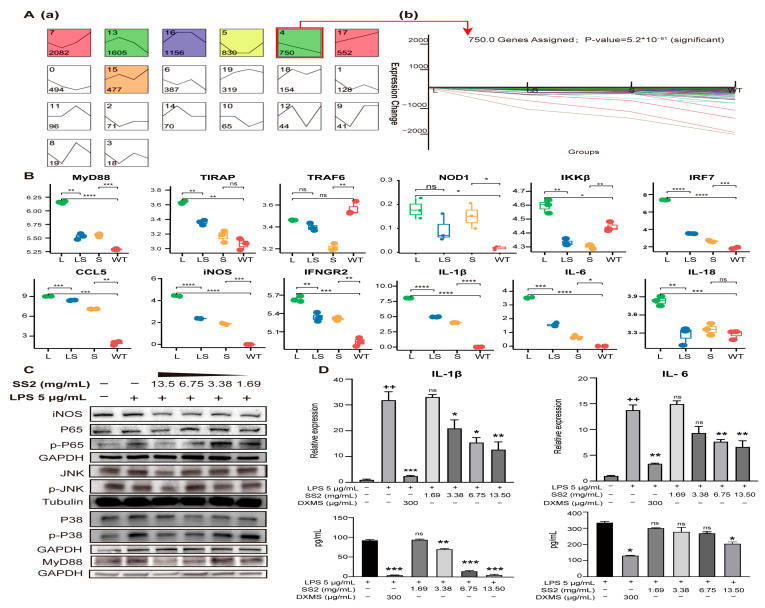
The gene expression tendency of MyD88/NF-κB signaling pathway. (**A**) Genes that showed the same expression tendency were clustered with the STEM cluster method. Profiles are ordered based on the number of genes assigned. The gene number of every cluster was marked (**a**). Typical gene expression changes details showed at (**b**). (**B**) Examples of genes whose expression was activated by the addition of LPS but repressed by sericin. Comparison of gene expression level among L, LS, S, WT. * *p* < 0.05, ** *p* < 0.01, *** *p* < 0.001, **** *p* < 0.0001, n = 3. (**C**) The changes in protein abundance of key nodes of the MyD88/NF-κB signaling pathway when sericin was added into the LPS-infected RAW264.7 cells. (**D**) The ELISA and qRT-PCR reports of IL-1β and IL-6. + Relative to no-treated cells. * Relative to LPS-treated cells. ++ *p* < 0.01, * *p* < 0.05, ** *p* < 0.01, *** *p* < 0.001, n = 3.

**Table 1 ijms-24-00259-t001:** The primary characteristics of the five sericin samples.

No.	Material State	Color	Smell	pH *	Aerobic Bacterial Count (CFU/mL (g))
S1	Powder	Pale Yellow	Strong Smell	5.36	<10
S2	Liquid	Brown	Moderate Smell	5.6	<10
S3	Liquid	Brown	Strong Smell	5.44	<10
SS1	Powder	Pale yellow	Light Smell	4.8	<10
SS2	Powder	Pale yellow	Light Smell	4.53	<10

* Configured as a 13.5 mg/mL sericin solution, tested at 22 °C.

**Table 2 ijms-24-00259-t002:** The proteins identified by LC-MS/MS.

Protein Classification	Protein ID	Data Base	S1 (%)	S2 (%)	S3 (%)	SS1 (%)	SS2 (%)
Sericin 1	NP_001037506.1	NCBI	49.46274901	5.335532292	8.839730442	53.30999038	68.01922871
	XP_012552503.2	NCBI	0	0	0	5.771091599	3.146896395
	XP_004934087.2	NCBI	0	0	0	2.572240592	1.684852375
	BGIBMGA002689	Silkdb 2.0	15.31242542	0.017105123	0	31.98284251	21.86263104
Sericin 2	NP_001166287.1	NCBI	1.950921623	0.018734328	21.94072201	0.336459243	0.596049957
Enzymes	XP_021208623.1	NCBI	1.259668974	0	61.73346493	0.096624902	0.077723845
	BGIBMGA014024	Silkdb 2.0	0	0.004999576	0	0	0.004847294
Protease inhibitors	BGIBMGA009092	Silkdb 2.0	6.380469726	0	6.315789474	0	0
	NP_001139708.1	NCBI	0.764392908	0.150414115	0.227766897	0	0
Unannotated Proteins	BGIBMGA000013	Silkdb 2.0	23.46805554	93.68313431	0.913997563	4.734896657	4.115853659
	BGIBMGA005069	Silkdb 2.0	1.401316805	0.790080258	0.028528691	1.195854114	0.49191672
Sericin content ratio			66.72609605	5.371371743	30.78045245	93.97262433	94.71360853

**Table 3 ijms-24-00259-t003:** End point score of HET-CAM.

NO.	Score of 6 Replicates						Total Score	Results
5% SS1	0	0	0	1	0	1	2	Non-irritant
7.5% SS1	1	1	0	1	0	0	3	Non-irritant
10% SS1	1	1	1	2	1	0	6	Non-irritant
5% SS2	1	0	1	1	0	1	4	Non-irritant
7.5% SS2	1	0	1	0	1	1	4	Non-irritant
10% SS2	1	1	2	1	1	2	8	Non-irritant
Blank (Saline)	0	0	0	0	0	0	0	Non-irritant
0.1M NaOH	9	12	9	12	6	9	57	Severe irritant

## Data Availability

The datasets of RNA-seq are available in the NCBI BioProject database (ID. PRJNA900300) (https://www.ncbi.nlm. nih.gov/bioproject/, accessed on 1 May 2022).

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
