# Peer review of "Multi-Omics Integration to Reveal the Mechanism of Sericin Inhibiting LPS-Induced Inflammation"

_ijms, 2022, doi:10.3390/ijms24010259_

Round 1

Reviewer 1 Report

In the presented manuscript the effec of Sericin on inflammation process has been observed. 

 I have some comments and questions:

1. in line 107, the phrase",respecively" is missing.

2. No spaces before each quote

3. Figure 3. G,H,And why are the S2 effects different from the others? What could be the reason?

4. Figure 3. G (a) axis description is illegible and can be misleading.

5. Figure 3. G,H,I What concentrations of Sericin and LPS were used?

6. What method was used to calculate the mRNA level? If the results were normalized to the level of GAPDH, they should not be close to 1? Why are the untreated LPS and Sericin controls missing from the charts?

7. Figure 4 and 5 are illegible. my advice is to split these Figures into 2 separate Figures.

8. Figure 5. Are all Western blots normalized to one GAPDH membrane? If so, this is an error.

9. Were the phosphorylated forms of proteins tested on the same membranes as the non-phosphorylated forms? If so (as indicated by the analyzes of the membranes), by what method were the membranes recovered?

10. Are WB analyzes from only one replicate?

Figure 5. E - Which Sericin samples were tested and at what concentration? this should be supplemented.

11. Figure 5 F- Why was SS2 sample selected for testing and not other samples?

Author Response

We would like to thank the reviewers for their valuable feedback. We use it to improve the quality of the manuscript. Please refer to the attachment.

Reviewer 2 Report

In this manuscript, the authors first determined the protein composition of sericin extracts and then studied the anti-inflammatory mechanism of the protein. However, despite the interesting work I must point out some revisions:

1)      The authors selected the SS2 sample for their cellular and molecular investigations. Please specify this in the abstract and results (including figures)

2)      line 49: it is preferable to use the word researchers rather than scholars

3)      in the introduction section the description of the results is too long (see line from 92 to 101)

4)      In the materials and methods section the description of the western blot analyzes is missing. Please add.

5)       I require a major revision of the English. This would make reading the discussion more fluid.

Author Response

(The authors gave the same response as above.)

Round 2

Reviewer 1 Report

Figure 3. G (a), H (a) and I (a) axis description is still illegible and can be misleading. You need to improve it before acceptance. 

Author Response

Thank you for your feedback.

We have humbly made changes to Figure 3G. We sincerely hope the readability of the figures is improved.

Thank you for your hard work during the revision of the manuscript.